# Asymmetrizing an icosahedral virus capsid by hierarchical assembly of subunits with designed asymmetry

Zhongchao Zhao [1,5], Joseph Che-Yen Wang[1,2,3], Mi Zhang[4], Nicholas A. Lyktey[4], Martin F. Jarrold[4], Stephen C. Jacobson [4] & Adam Zlotnick [1✉]

Symmetrical protein complexes are ubiquitous in biology. Many have been re-engineered for chemical and medical applications. Viral capsids and their assembly are frequent platforms for these investigations. A means to create asymmetric capsids may expand applications. Here, starting with homodimeric Hepatitis B Virus capsid protein, we develop a heterodimer, design a hierarchical assembly pathway, and produce asymmetric capsids. In the heterodimer, the two halves have different growth potentials and assemble into hexamers. These preformed hexamers can nucleate co-assembly with other dimers, leading to Janus-like capsids with a small discrete hexamer patch. We can remove the patch specifically and observe asymmetric holey capsids by cryo-EM reconstruction. The resulting hole in the surface can be refilled with fluorescently labeled dimers to regenerate an intact capsid. In this study, we show how an asymmetric subunit can be used to generate an asymmetric particle, creating the potential for a capsid with different surface chemistries.

[1] Molecular and Cellular Biochemistry Department, Indiana University, Bloomington, IN 47405, USA. [2] Indiana University Electron Microscopy Center, Indiana University, Bloomington, IN 47405, USA. [3] Department of Microbiology and Immunology, Pennsylvania State University College of Medicine, Hershey, PA 17033, USA. [4] Department of Chemistry, Indiana University, Bloomington, IN 47405, USA. [5] Present address: Department of Nanoengineering, University of California San Diego, La Jolla, CA 92039, USA. ✉email: azlotnic@indiana.edu

Symmetrical supramolecular protein complexes are ubiquitous in natural biological systems to compartmentalize and execute complex functions[1–5]. Many groups have exploited natural systems to develop nanotechnologies for drug delivery, energy transport, and information storage[6–17]. Symmetrical viral capsids and their assembly are frequent platforms for these investigations[18–22]. Many viral capsids have icosahedral symmetry and are assembled from a single symmetrical building block[23–26]. The simplicity of viral capsids provides many advantages. However, symmetrical subunits offer little or no opportunity to control the reaction and to incorporate specific asymmetric features. This shortcoming may limit development of applications that need conditional stops for information insertion and cargo loading. Engineering a controlled assembly pathway with designed pauses is a strategy to overcome this drawback and support hierarchical assembly of a capsid.

Although capsid assembly has nucleation and elongation phases, there are no discrete stopping points for manipulation and modification of specific intermediates[27–29]. Here we used Hepatitis B Virus (HBV) capsid assembly as a model system and demonstrated an effective approach to break the spontaneous assembly into addressable steps[28,30,31]. HBV capsid dimeric subunits can assemble two species of capsids, $T = 3$ capsid with 90 dimers and $T = 4$ capsid with 120 dimers. $T = 4$ capsid is the predominant species and the capsid within infectious virions[32]. HBV dimers interact when the end of the contact helix of one dimer fits into a groove formed by the contact helix of a subunit from an adjacent dimer (note the hexamer in Fig. 1). Of course, a dimer has two contact helices at either end[23,33,34]. In concept, both monomers can be engineered differently, leading to a heterodimer for which each monomer can only assemble with other dimers in response to a specific condition. One extreme example would be a heterodimer with an assembly-active monomer and an assembly-incompetent monomer[35–37].

In this work, we introduce a platform and some basic manipulations for controlled assembly. We designed a heterodimer and have used it to generate small complexes, which, in turn, can nucleate assembly of a capsid. The co-assembled capsid has two discrete and addressable patches, a heterodimer patch and a homodimer patch, analogous to a Janus particle. By virtue of a distinct hydrophobic patch, Janus particles can be driven to assemble into larger supramolecular structures. Here, the chemically distinct nature of the two patches give us the ability to control further modification, disassembly, and reassembly.

## Results

**Design of a pathway for asymmetric capsid assembly.** To create an asymmetric assembly pathway with asymmetric capsids requires a means to initiate assembly with a specific complex and a means of halting and possibly restarting assembly at a specific juncture. As an example, we designed an asymmetric assembly pathway from asymmetric subunits to asymmetric holey capsids, and eventually to asymmetric $T = 4$ capsids (Fig. 1). With this designed pathway, we created stages in the reaction, which yield opportunities for modifications and engineering.

To achieve asymmetric assembly, we modified the 149-residue HBV capsid protein assembly domain, Cp149. We created a heterodimeric subunit, $Cp149_{His}Cp149_{Y132A}$ (Fig. 2a), which possesses functionalities on each monomer, one encoding a programable assembly function and the other encoding a conditional stop. The assembly-active monomer can assemble in response to ionic strength, such as wild-type dimer, and is also sensitive to $Ni^{2+}$ due to the addition of a His-Tag[38]. The conditional stop monomer carries the assembly-incompetent mutation Y132A, which inhibits assembly due to loss of hydrophobic surface for proper subunit interactions; however, it can still co-assemble with wild-type dimers into labile capsids[35–37].

In our engineered assembly path (Fig. 1), the asymmetric heterodimers assemble in response to $Ni^{2+}$ but will stop after forming an initial complex, a hexamer in this diagram, due to the influence of Y132A. Hexamers then can be used as nuclei to co-assemble with a second species of dimer, in response to high ionic

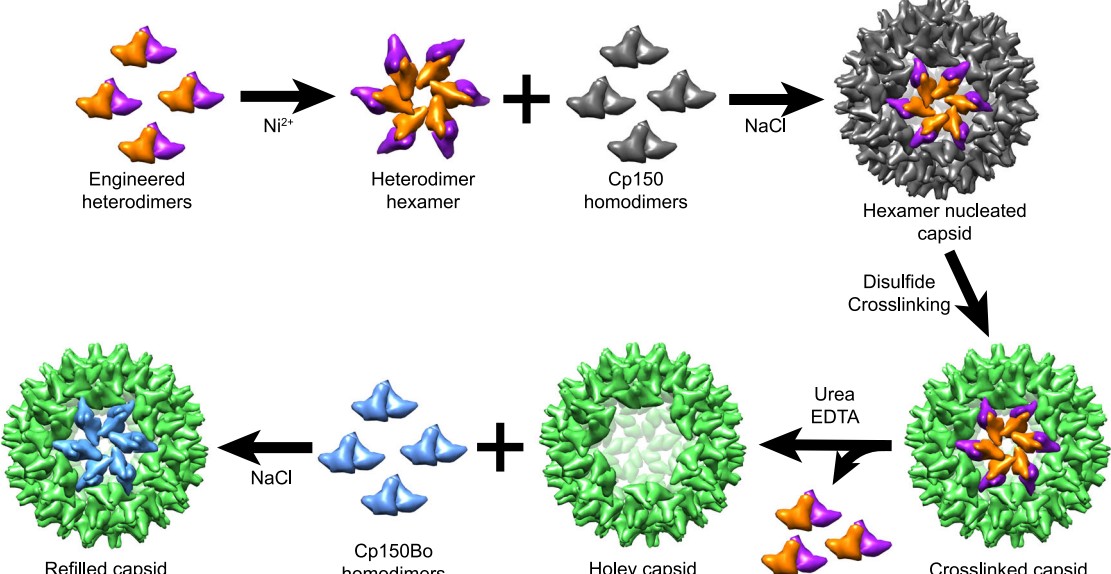

**Fig. 1 Schematic of the hierarchical assembly of an HBV capsid analog starting with an asymmetric dimeric subunit.** Engineered HBV heterodimers (orange and purple) assemble into hexamers by addition of $Ni^{2+}$. The hexamer model in this figure provides a proposed representation of the interaction between dimers. Hexamers nucleate co-assembly with Cp150 homodimers (gray) into capsids. After crosslinking the Cp150 surface (now green), co-assembled capsids are disassembled into holey capsids by removing the patch corresponding to the nucleating hexamer. Holey capsids then can be surface-refilled into compete capsids with the addition of a new species of homodimer (blue). It is noteworthy that the co-assembled capsid and refilled capsid have a defined patch, allowing display of a unique surface chemistry. Also, the opening in the holey capsid offers the opportunity to add a cargo to the capsid.

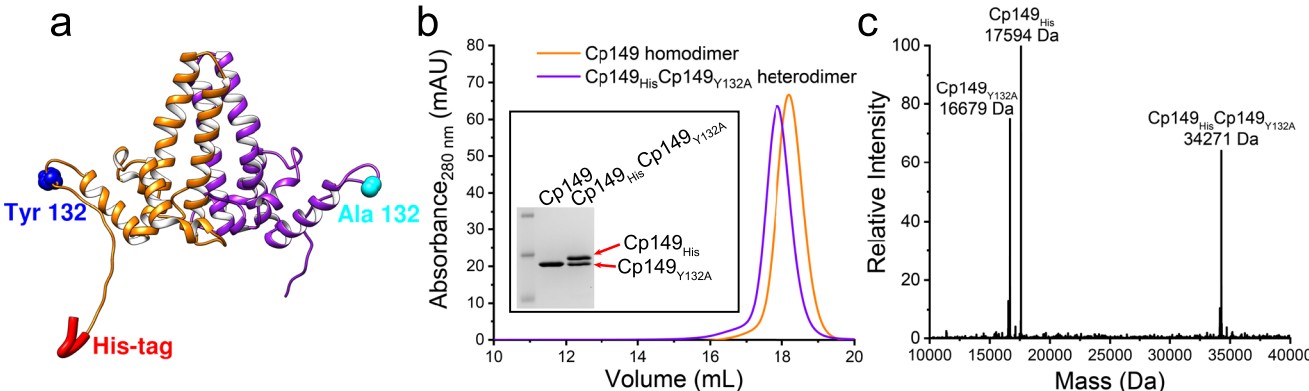

**Fig. 2 Design and purification of the engineered heterodimer, Cp149$_{His}$Cp149$_{Y132A}$. a** A model of the modified heterodimer based on 1QGT. The contact helix and subsequent loops, with residue 132, extend to the right and left of the base of the dimer. A six-histidine His-Tag is attached to the C terminus of one monomer (orange) and an assembly-incompetent mutation (Y132A) is present on the other monomer (purple). **b** Purified heterodimer, Cp149$_{His}$Cp149$_{Y132A}$, shows a single peak by SEC (purple) and elutes similarly to purified Cp149 homodimers (orange). SDS-PAGE shows a single band for Cp149 homodimers, whereas Cp149$_{His}$Cp149$_{Y132A}$ shows two equally dense bands corresponding to the two different monomers (inset). **c** Native mass spectrometry of Cp149$_{His}$Cp149$_{Y132A}$ shows three species corresponding to the different monomers and the disulfide crosslinked heterodimer. No MS evidence was found for the Cp149$_{His}$ homodimer or Cp149$_{Y132A}$ homodimer.

strength. The resulting hybrid capsids have two distinct patches, the hexamer nucleus, and the homodimer component. To further manipulate the structure, we differentially stabilize the patches, which can be accomplished by using a homodimer that can spontaneously form crosslinks[39,40]. Here we used Cp150 homodimers. As a Cp149 variant, Cp150 incorporates a C-terminal cysteine that clusters at fivefold and quasi-sixfold vertices resulting in disulfide crosslinks[33,39,40]. Crosslinked regions in hybrid capsids are stable under low ionic strength and urea treatment. As the heterodimer lacks cysteine 150 and fails to crosslink, heterodimer patches can be removed specifically, leaving crosslinked asymmetric holey capsids. Such holey capsids then can be further modified and/or the surface can be refilled with new subunits to generate symmetric or asymmetric, un-holey $T = 4$ complete capsids.

In summary, we have a series of essentially orthogonal reactions that allow us to engineer features into each step of a hierarchical assembly: (i) nucleate, (ii) elongate to create a body, (iii) crosslink capsid body, (iv) remove nucleus, and (v) refill the surface.

**Purification and characterization of heterodimer Cp149$_{His}$Cp149$_{Y132A}$.** To generate the asymmetric heterodimer Cp149$_{His}$Cp149$_{Y132A}$, we designed a bicistronic expression plasmid (Supplementary Fig. 1). This approach has been used to split a single monomer into two segments, called SplitCore, which can still dimerize, to incorporate oversized proteins for vaccine development[41]. The bicistronic plasmid carries a single promoter followed by a gene for each monomer, Cp149$_{His}$ and Cp149$_{Y132A}$. Each gene has its own ribosome-binding site. Following *Escherichia coli* expression, Cp149$_{His}$Cp149$_{Y132A}$ was purified as a dimer and characterized. Purification included size-exclusion chromatography (SEC) to remove assembly-competent Cp149$_{His}$ homodimer and Ni-NTA affinity chromatography to isolate His-tagged heterodimer from Cp149$_{Y132A}$ homodimer. The yield of purified protein was ~50 mg per liter of Luria broth (LB). Cp149$_{His}$Cp149$_{Y132A}$ eluted as a single peak on SEC at the same position as homodimer Cp149 (Fig. 2b). Cp149$_{His}$Cp149$_{Y132A}$ was resolved on SDS-polyacrylamide gel electrophoresis (PAGE), showing approximately equal amounts of two bands corresponding to the Y132A and the His-tag monomers (Fig. 2b, inset). Native mass spectrometry (MS) of heterodimer Cp149$_{His}$Cp149$_{Y132A}$ showed three peaks corresponding to Cp149$_{His}$ monomer, Cp149$_{Y132A}$

monomer, and heterodimer comprising Cp149$_{His}$ and Cp149$_{Y132A}$ (Fig. 2c). There was no evidence of either Cp149$_{His}$ or Cp149$_{Y132A}$ homodimers in the MS analysis.

Efficient expression of highly purified heterodimer is an important step towards asymmetric assembly in this study. An asymmetric dimer may also be useful for vaccine development and other applications[41–44]. The splitcore system allowed incorporation of large inserts but still generated a symmetric dimer[41]. An HBV tandem dimer comprising two monomers linked by a short peptide was developed[45,46]. However, in our hands, we observed the tandem dimer had low yield and generated many aggregates, which suggested that the linkage between monomers may have led to misfolding. The bicistronic heterodimer expression system retained monomer integrity, while still providing the opportunity to modify each monomer independently.

**Heterodimer Cp149$_{His}$Cp149$_{Y132A}$ assembles to form a hexamer.** As heterodimer carries the assembly-incompetent mutation Y132A[35–37], we first compared the assembly induced by high ionic strength (300 mM NaCl) of Cp149$_{His}$Cp149$_{Y132A}$ and wild-type homodimer Cp149. As anticipated, Cp149$_{His}$Cp149$_{Y132A}$ failed to assemble capsids and no stable intermediates were isolated under conditions where Cp149 assembled readily (Supplementary Fig. 2).

We then tested assembly of Cp149$_{His}$Cp149$_{Y132A}$ in response to Ni$^{2+}$ at low ionic strength, taking advantage of the His-tag without involving ionic strength-driven assembly (Fig. 3). We typically used 10 μM heterodimer and 100 μM NiCl$_2$ for these reactions. We observed by SEC that Cp149$_{His}$Cp149$_{Y132A}$ assembles into heterogeneous complexes that are larger than dimers. Using charge detection MS (CDMS), a single-molecule native MS technique capable of resolving the masses of complex mixtures[47], we found that major species were free dimers, dimers of dimers, and hexamers with some putative 11-dimer double hexamers (calculated masses 34.3, 68.6, 205.6, and 377.0 kDa, respectively, with the retention of water and salts contributing to the offset of peaks) (Fig. 3b). The larger complexes were isolated by SEC and their presence confirmed by negative-stain electron microscopy (EM) (Supplementary Fig. 3). Particles from micrographs were semi-manually selected and subjected to two-dimensional (2D) classification, resulting in top and side views of hexamers and double hexamers (Fig. 3c). Due to the limited number of particles, a density map of a double hexamer was not

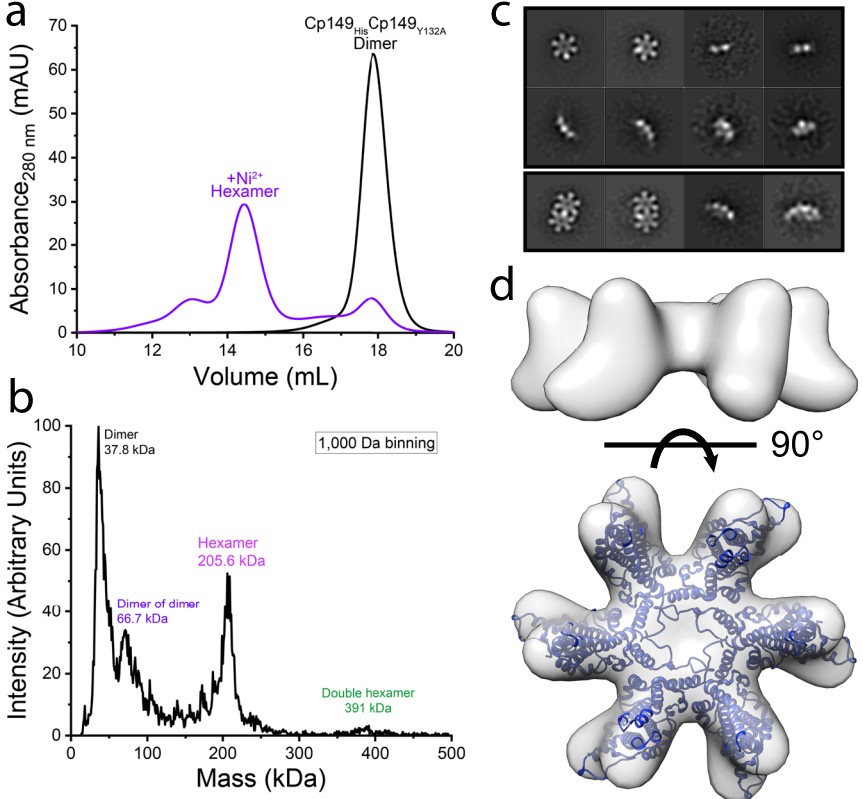

**Fig. 3 Characterization of heterodimer hexamers. a** SEC shows that with the addition of Ni$^{2+}$, Cp149$_{His}$Cp149$_{Y132A}$ heterodimers form larger species and elute earlier (purple) than heterodimers (black). **b** CDMS demonstrates that Ni$^{2+}$ induces formation of hexamers and larger species. **c** Class average of negative-stain EM of purified hexamers shows top and side views of hexamers. Double hexamers are also observed. Raw data are shown in Supplementary Fig. 3. **d** A 17 Å resolution hexamer density map reconstructed from negative-stain EM (side and top views) (EMD-22133) matches well with a hexamer model isolated from an HBV capsid (1QGT).

reconstructed. The top eight classes were selected for reconstructing a three-dimensional (3D) density map that reached 17 Å resolution. The resulting density map confirmed assembly of capsid-like hexamers by Cp149$_{His}$Cp149$_{Y132A}$ (Fig. 3d). A hexamer molecular model, isolated from an HBV capsid[48], fit well into the density. Thus, unlike symmetric homodimers, which assemble into capsids with very low concentrations of intermediates, asymmetric Cp149$_{His}$Cp149$_{Y132A}$ heterodimers assemble to hexamers or double hexamers and then assembly stops.

**Cp149$_{His}$Cp149$_{Y132A}$ hexamers co-assemble with homodimers.** Although the Y132A mutant is assembly incompetent on its own, it can co-assemble with Cp149[35–37]. We reasoned that the hexameric complex that assembled in response to Ni$^{2+}$ could nucleate further assembly, because each incoming dimer will make two contacts to the nucleating hexamer, only one of which was compromised by the Y132A mutation. Thus, the Y132A may weaken the initial steps of assembly but will not prevent it. To test this hypothesis, we co-assembled preformed hexamers with homodimers, in effect making a Janus particle with a nucleus patch and a much larger homodimer patch[49,50]. Here we chose Cp150 homodimers, because the resulting capsids can crosslink through C150 disulfide formation yielding capsids that are stable even in 5 M urea[51]. In the co-assembly reaction, the homodimer component would be sturdy enough to allow subsequent removal of the heterodimer hexamers.

We observed although heterodimer polymerization stops at hexamers, hexamers can nucleate ionic strength-driven (300 mM NaCl) assembly of Cp150 homodimers, forming hybrid capsids; adding heterodimer hexamers to an assembly reaction promoted

assembly and decreased the apparent pseudo-critical concentration of assembly (Fig. 4a). As a control, we observed that Ni$^{2+}$ at a concentration of 100 µM had no measurable effect on Cp150 assembly (Supplementary Fig. 4). Free Cp149$_{His}$Cp149$_{Y132A}$ heterodimers can co-assemble with Cp150 to form morphologically normal capsids, but there is no evidence that the different classes of subunit segregate (Supplementary Figs. 4 and 5).

Structural and model studies have suggested that a trimer of dimers acts as the nucleus in normal HBV capsid assembly[27,31,36]. Here we showed that artificially created hexamers can also function as nuclei to promote assembly. It suggests that capsid assembly can progress by various pathways that might arise from different types of nuclei. Manipulation of nuclei formation may change the course of assembly.

**Resection of hybrid capsids to holey capsids.** Previously, we attempted to generate holey capsids by removing modified subunits that had been incorporated stochastically; we were unable to identify regular, contiguous patches by this method[51]. However, our Janus capsids have a heterodimer hexamer patch in a body of crosslinked Cp150. We hypothesized that removing the heterodimer hexamers will leave Cp150 holey capsids intact due to their exceptional stability. To test this hypothesis, we purified hybrid capsids from a co-assembly reaction and removed heterodimer hexamers with a cocktail of 100 µM EDTA and 3 M urea. EDTA will disrupt the Ni$^{2+}$-mediated interaction between His-Tags. Urea at 3 M weakens Cp–Cp interactions without unfolding dimer structures, allowing the heterodimer Cp149$_{His}$Cp149$_{Y132A}$ dimers to dissociate from the hybrid capsids[52]. We observed that EDTA and urea treatment decreased the amount of Cp149$_{His}$

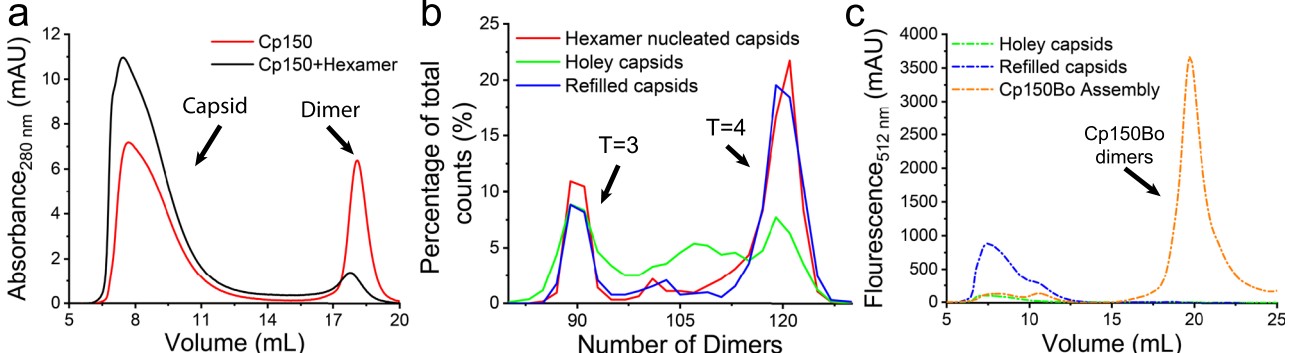

**Fig. 4 Generating holey capsids through co-assembly and disassembly, and reassembling holey capsids. a** SEC shows that heterodimer hexamers co-assemble with Cp150 homodimers (black) to generate more capsids than Cp150 homodimer itself (red). **b** RPS analysis of co-assembled capsids, holey capsids, and surface-refilled capsids. Purified co-assembled capsids have two major species, $T = 3$ and $T = 4$ capsids (red line). After the disassembly process, holey capsids are generated from $T = 4$ capsids resulting in a heterogenous array of incomplete holey capsids (green line). With the addition of Cp150Bo homodimer, the surfaces of holey capsids are refilled back to $T = 4$ capsids (blue line). **c** SEC with a fluorescence detector shows that Cp150Bo co-elute with surface-refilled capsids. Chromatographs are shown for holey capsids (green), surface-refilled capsids (blue), and, as a control, Cp150Bo capsids (orange). It is noteworthy that the BODIPY fluorescence in the capsids of pure Cp150Bo is quenched.

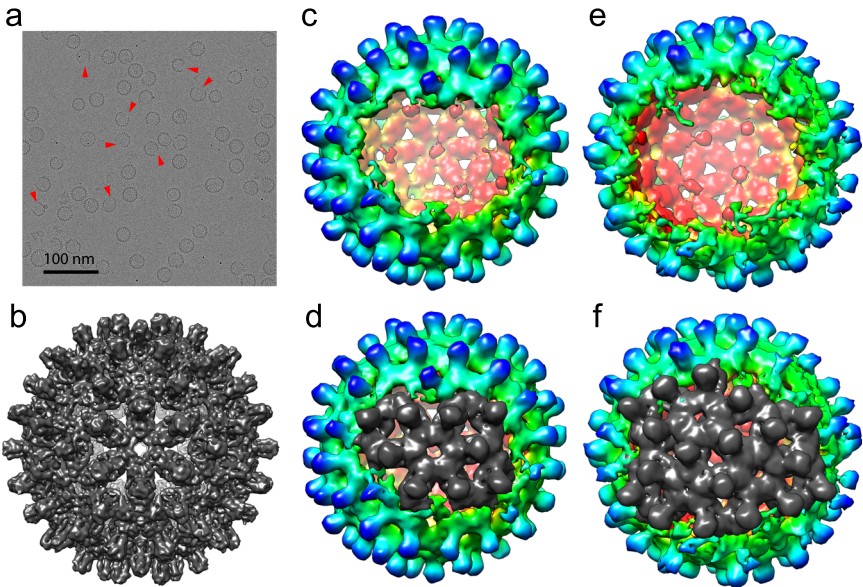

**Fig. 5 Cryo-EM characterization of holey capsids. a** A cryo-EM micrograph of a holey capsid sample. Several obvious holey capsids are pointed out by red arrows. **b**, **c**, **e** Reconstructions yielded a 120-dimer complete capsid (EMD-22132) (**b**) and two species of holey capsids (**c** EMD-22134 and **e** EMD-22135). **d**, **f** Comparing holey to complete capsid maps shows that the capsid with the small hole (**c**, **d**) is missing density for nine dimers including one hexamer. Whereas the capsid with a bigger hole (**e**, **f**) is missing density for ~18 dimers including two hexamers and a pentamer. The partial subunits at the periphery of the holes suggests heterogeneity of hole size and mobility of peripheral dimers. For these figures, capsid density is contoured at $1.3\sigma$. Holey capsids are colored by radius for contrast (blue 35 nm, green 32 nm, red 29 nm).

associated with the presumed holey capsids based on SDS-PAGE and led to altered elution of capsid-sized particles on SEC (Supplementary Fig. 6).

To validate the presence of holey capsids and not the selective dissociation of hexamer-enriched capsids, we analyzed purified holey capsids by resistive pulsive sensing (RPS)[53,54], which is a particularly powerful approach for working with low protein concentrations in the presence of non-volatile buffers. In RPS, a particle with lower conductivity displaces a proportional amount of electrolyte from a nanopore, resulting in a reduction of the electric current that is proportional to the volume of the particle (Supplementary Figs. 7 and 8). We calibrated the nanofluidic device with Cp150 capsid standards (Supplementary Fig. 9) and used RPS to determine the particle-size distributions of hexamer-nucleated, holey, and surface refilled capsids (Fig. 4b). As seen in

Fig. 4b, we found that EDTA and urea reduced the proportion of 120-dimer $T = 4$ capsids, had little effect on the proportion of 90-dimer $T = 3$ particles, and introduced a new heterogeneous mixture of incomplete, stable capsids (Fig. 4b, green line). The remaining $T = 4$ and $T = 3$ capsids appear to be intact and may be assembled exclusively from Cp150 homodimers.

**Cryo-EM reconstruction of holey capsids.** To unambiguously confirm that asymmetric holey capsids had been created, we characterized the species in a resection reaction with cryo-EM. As predicted from our RPS results, raw micrographs of holey capsids showed a mixture of compete capsids and capsids with open edges (Fig. 5a, red arrows). Next, 41,057 particles were selected and processed for 3D image reconstruction. Through the whole

reconstruction process, C1 symmetry was applied to avoid losing unique features that would have been overwritten if we had used icosahedral symmetry averaging[55].

After 2D classification, the dataset was divided into two sets, one for complete capsids (23,478 particles) and the other for asymmetric holey capsids (17,579 particles). The dataset for complete capsids yielded a 6.2 Å resolution density map showing an apparently symmetrical 120-dimer capsid, though only C1 symmetry was applied (Fig. 5b). We note that some capsids included in the complete capsid dataset may have had holes that were obscured by particle orientation. The holey capsid dataset was further divided into two sets by 3D classification, based on the size of the hole. In the end, 9697 particles were used to reconstruct a 10.8 Å resolution density map for a holey capsid that was missing around a hexamer of dimers (Fig. 5c). The remaining 7882 particles were reconstructed to yield a 12.9 Å resolution density map for a holey capsid missing approximately two hexamers (Fig. 5e). The two holey capsid density maps never reached the same resolution as the 120-dimer capsid density map. We believe that might be due to the use of fewer particles, particle heterogeneity, and potentially the greater flexibility of holey capsids.

To quantify the number of missing dimers on both reconstructions, we overlaid density maps for holey capsids at $1.3\sigma$ contour level on the density map for the 120-dimer capsid density map. For the capsid with the smaller hole, density was missing for ~9 dimers centered around a hexamer (Fig. 5d). For the capsid with the bigger hole, ~18 dimers were missing from a hole that could be filled by a double hexamer and a pentamer (Fig. 5f). Initially, we suggested that during co-assembly, hexamers and double hexamers nucleated capsid formation. However, based on RPS, there was no clear peak for holey capsids that lost exactly a hexamer or a double hexamer during capsid resection. Careful examination of both holey capsid density maps showed degraded density close to edge of the hole, suggesting that particles used for reconstruction were not homogeneous. This heterogeneity could arise for multiple reasons: (i) additional heterodimers could have associated with the built-in hexamer or the double hexamer during the co-assembly reaction, which would result in heterogeneity at edge of the hole, and (ii) although the majority of the Cp150 homodimers crosslinked, dimers on the edge of the hole may not have crosslinked and were thus labile when exposed by resection. The heterogeneity of the holes observed by cryo-EM matches well with the apparent heterogeneity defined by RPS of holey capsids (Fig. 4b).

**The surface of holey capsids can be refilled to create asymmetric, un-holey capsids.** Early attempts to characterize assembly pathways showed intermediates with sizes similar to holey capsid species that could proceed to assemble. However, we were unable to structurally characterize and isolate them, which may be due to their rarity and instability[47]. We reasoned that our holey capsids can also proceed to assemble to 120-dimer un-holey capsids, making them useful for containing cargo and for displaying surface features that were not part of the original nucleus or the capsid body. In effect, holey capsids are models of on-path assembly intermediates[47].

To test our ability to refill the surface of the holey capsids, we introduced a fluorescent homodimer, Cp150Bo, which has a BODIPY fluorophore attached to residue cysteine 150 of Cp150[39]. After the reassembly reaction, using RPS, we observed that the amount of heterogenous holey capsids decreased nearly to the baseline and the amount of 120-dimer capsids, indistinguishable from control $T = 4$ capsids, increased (Fig. 4b, blue line). This result indicated that holey capsids are assembly-active

polymers to which Cp150Bo homodimers can be added. The appearance of absorbance at 504 nm (Supplementary Fig. 10b) and bright fluorescence at 512 nm for BODIPY fluorophore in chromatographic separations of un-holey capsids confirmed the addition of Cp150Bo (Fig. 4c, blue). As a control, capsids of pure Cp150Bo are dimly fluorescent (Fig. 4c, orange), because BODIPY fluorophores are close to each other in the capsid interior, leading to fluorescence quenching[48]. In surface-refilled holey capsids, there are no adjacent BODIPY fluorophores on the edges of holes, leaving fluorophores unquenched. We were able to quantify the amount of Cp150Bo eluting with capsid by comparing the absorbance for BODIPY per capsid between the surface-refilled holey capsids and Cp150Bo capsid (Supplementary Fig. 10), we calculated the Abs504/Abs280 ratio of both samples, we estimated that ~16 Cp150Bo dimers added on one holey capsid. The number of dimers in the refilled surface aligned well with our RPS and cryo-EM estimation. Of course, this estimation does not take into account the possibility of some pure Cp150Bo capsids and of capsids that lack any Cp150Bo.

These results demonstrate that holey capsids can react with free dimers, such as Cp150Bo. This result shows that other modified homodimers, potentially carrying moieties can be incorporated into the holey capsids through the reassembly process.

## Discussion

Symmetric subunits often spontaneously assemble into symmetric capsids with no stops[26,31]. Here we showed a strategy to break the continuous HBV capsid assembly into a series of independent reactions. The key to this process is to design a heterodimer with distinct assembly properties programmed into each monomer. One monomer is assembly-active and can assemble in response to $Ni^{2+}$, whereas the other monomer is assembly-incompetent alone but can co-assemble with additional homodimers[35–37]. By manipulating assembly conditions with the designed heterodimer, a set of independent assembly reactions can be achieved. The different reactions yielded: hexamer, asymmetric co-assembled capsids, asymmetric holey capsids, and surface-refilled asymmetric un-holey capsids. Formation of the asymmetric capsids mimics the assembly of bacteriophage P22 where a portal complex nucleates capsid assembly[56]. However, in this case, the hexamer portals can be removed to create asymmetric holey capsids where the surface can be subsequently refilled with other engineered dimers. This process provides many opportunities to engineer asymmetric and symmetric capsids for a range of applications[49,50], e.g., loading cargos into holey capsids, attaching a chemistry to a small contiguous patch of dimers. We are effectively providing a path for making very small, 36 nm diameter, Janus particles with molecular uniformity.

Many viral capsids have subunits that can be asymmetrized, although it is easiest to see doing this with monomers and dimers[23–26,57]. We anticipate that our strategy to create an asymmetric assembly pathway and asymmetric capsids can be applied to other capsid-like systems. There are likely to be some fundamental requirements needed to thermodynamically separate the individual steps of the reactions. First, the self-assembly system must be reversible. Second, it must be possible to create an asymmetric subunit where at least one face is able to conditionally stop assembly under conditions where the other face assembles. In HBV, the Y132A mutation provided an additional barrier to assembly that might have proceeded once a nucleus is formed, even at low ionic strength. Third, it must be possible to overcome the conditional stop. In HBV, we depended on the multivalent binding sites afforded by the hexameric nucleus and the relatively high association energy for subunit addition afforded by high ionic strength. Finally, there must be a substantial energy gradient

between the nucleus and the growing capsid that can be manipulated to change which domain is more stable. In low ionic strength and moderate $Ni^{2+}$ concentration, the hexameric nucleus is more stable than capsid. After disulfide crosslinking and in the presence of EDTA and urea, the Cp150 capsid is far more stable than the hexamer. Engineering a crosslinkable His-tag and a crosslinkable cysteine into a location near a vertex would seem a good starting strategy. We note the use of elastin-like peptides in Cowpea Chlorotic mottle virus capsid protein as an example of a capsid with two modes of regulatable assembly[58].

Asymmetric capsids can figuratively and literally open a window to create organized, multi-functionalized particles. Here we provide a platform and a strategy, adaptable to other platforms, for generating asymmetric particles. An advantage of capsid-like particles is their ability to display clusters of an epitope or a receptor-binding tag[59,60]; assembly of an asymmetric particle has the potential to create patches of two different ligands to take advantage of multivalent display and high-avidity binding. Asymmetry can be leveraged to create Janus particles that, e.g., adsorb to each other to make supramolecular structures or adsorb to a surface to generate a molecular monolayer[49,61,62]. A holey capsid is a unique structure that allows unique access to the sequestered interior of the particle as well as release of its contents providing flexibility for a capsid-like structures that have already been repurposed as a container for delivery or as vessel for chemistry[18,63–65].

## Methods

**Design of the bicistronic expression plasmid pET11a-Cp149$_{His}$Cp149$_{Y132A}$.**
Two individual expression plasmids, pET11a-Cp149$_{His}$ and pET11a-Cp149$_{Y132A}$ for protein Cp149$_{His}$ and Cp149$_{Y132A}$, respectively, were synthesized by GeneU-niveral. To guarantee the downstream construction for heterodimer plasmid, the expression sequences for Cp149$_{His}$ and Cp149$_{Y132A}$ were separately inserted between restriction sites for enzyme NdeI and NheI on two naked pET11a plasmids. Plasmid pET11a-Cp149$_{His}$ was digested with NheI and BamHI, and plasmid pET11a-Cp149$_{Y132A}$ was digested with XbaI and BamHI. Fragments carrying expression genes were recovered with the QIAquick Gel Extraction Kit. As XbaI and NheI share the same overhang nucleosides, CTAG, both fragments were mixed and ligated together leading to the pET11a plasmid carrying a bicistronic expression gene for Cp149$_{His}$Cp149$_{Y132A}$.

**Protein expression of homodimers Cp149, Cp150 and heterodimer Cp149$_{His}$Cp149$_{Y132A}$.** Cp149 and Cp150 were purified from an *E. coli* expression system. Capsids from lysate were precipitated by ammonium sulfate, followed by SEC, dissociation of capsid to dimers by 3 M urea, and a second round of SEC to isolate dimers[66]. For heterodimer Cp149$_{His}$Cp149$_{Y132A}$, the bicistronic expression plasmid, pET11a-Cp149$_{His}$Cp149$_{Y132A}$, was transformed into *E. coli* BL21(DE3) for protein expression. The purification protocol was modified based on the previously published protocol for Cp149$_{Y132A}$[37], Cp150[40], and Cp149D78S[66]. Simply after transformation, single colonies were inoculated in 5 mL LB media with 100 μg/mL carbenicillin at 37 °C overnight, which was then transferred into 250 mL Terrific broth (TB) with 100 μg/mL carbenicillin for 16 h at 37 °C without isopropyl β-D-1-thiogalactopyranoside induction. Twenty-six grams of cells were collected out of 1.75 L TB media by centrifugation. All following steps were finished either on ice or at 4 °C, and all buffer solutions were pre-equilibrated at 4 °C. Cell paste was resuspended with a similar lysis buffer except replacing 50 mM TRIS pH 7.5 by 50 mM HEPES pH 7.5 and lysed by sonication as published for Cp150 purification[40]. After sonication and ammonium sulfate precipitation, the precipitated protein pellet was resuspended in 40 mL 50 mM NaHCO$_3$ pH 9, 2 mM dithiothreitol (DTT) (Buffer X), then loaded onto a hand packed XK 50/100 column with Sephacryl S300 resin (GE Healthcare) and eluted overnight with Buffer X. Heterodimer Cp149$_{His6}$Cp149$_{Y132A}$ is supposed to be an assembly-incompetent dimer with the Y132A mutation and the two monomers, Cp149$_{His6}$ and Cp149$_{Y132A}$, and should have 1 to 1 ration. To exclusively collected heterodimers, fractions were evaluated by SDS-PAGE and absorbance, and fractions meeting the two conditions were collected. Pooled fractions were further purified as Cp149Y132A with a HiTrap Q HP column (GE Healthcare)[37] to remove junk proteins. Pooled fractions from the HiTrap Q step were precipitated using 20% w/v ammonium sulfate, then resuspended in 20 mL 20 mM TRIS-HCl, 50 mM NaCl, 5 mM Imidazole, 2 mM DTT (Buffer A), and loaded on a HisTrap HP column (GE Healthcare). Heterodimer Cp149$_{His6}$Cp149$_{Y132A}$ bound to the column was eluted with 43% 20 mM TRIS-HCl, 50 mM NaCl, 500 mM Imidazole, 2 mM DTT (Buffer B) after a linear gradient test. Based on SDS-PAGE evaluation, fractions with pure proteins were collected and dialyzed into Buffer X, stored at

−80 °C. In all experiments, all proteins were dialyzed into 50 mM HEPES pH 7.5 prior to use. All protein concentrations were calculated using the extinction coefficient ($ε_{280} = 60,900$ M$^{-1}$ cm$^{-1}$) for HBV Cp149 homodimer.

**SDS-PAGE and mass spectrometry**. To examine the protein purity and content, heterodimer Cp149$_{His}$Cp149$_{Y132A}$ was first diluted to 20 μM. Twenty microliters of Cp149 or Cp149$_{His6}$Cp149$_{Y132A}$ was mixed with 20 μL 4× protein SDS-PAGE loading buffer without β-mercaptoethanol and boiled for 8 min at 95 °C. Five microliters of each reaction was loaded onto a 6–16% SDS-PAGE for analysis. Freshly prepared Cp149$_{His6}$Cp149$_{Y132A}$ was also analyzed on a Waters Synapt G2-S mass spectrometer to determine the purity and contents.

For co-assembled capsids and holey capsids, 15 μL of each sample at 6 μM was mixed with 15 μL 4× protein SDS-PAGE loading buffer with β-mercaptoethanol and boiled for 8 min at 95 °C. Fifteen microliters of each sample was loaded onto a 6–16% SDS-PAGE for analysis.

**Size-exclusion chromatography**. All the analytical experiments were carried out using a Superose 6 10/300 GL column (GE Healthcare) mounted on an high-performaance liquid chromatography (HPLC) system (Shimadzu) with a 0.5 mL/min flow rate. To collect fractions for further characterization, all reactions were loaded onto the same column mounted on a GE AKTApurier FPLC. For different experiments, the column was equilibrated in a corresponding buffer. For heterodimer hexamer characterization, the column was equilibrated and run with 50 mM HEPES pH 7.5. For Cp150 and hexamer co-assembly reactions, the column was run with 300 mM NaCl, 50 mM HEPES pH 7.5. For the disassembled holey capsids, the column was run with 150 mM NaCl, 50 mM HEPES pH 7.5. For the refilling reactions, the column was run with 300 mM NaCl, 50 mM HEPES pH 7.5.

**Assembly of hexamer, co-assembly, disassembly, and holey capsids refilling**. All concentrations mentioned are final concentrations. To assemble heterodimer hexamers, heterodimer Cp149$_{His}$Cp149$_{Y132A}$ was mixed with NiCl$_2$ in 50 mM HEPES pH 7.5 buffer with a 1 : 10 ratio of Cp149$_{His}$Cp149$_{Y132A}$ to NiCl$_2$ and incubated at 23 °C for 1 h. To use hexamers as nuclei for Cp150 co-assembly, a 4 : 1 ratio of Cp150 to Cp149$_{His}$Cp149$_{Y132A}$ was mixed with the pre-assembled hexamers, leading to a 4 : 1 : 10 ration of Cp150 : Cp149$_{His}$Cp149$_{Y132A}$ : NiCl$_2$ in the mixture, and incubated for 10 min at 23 °C. Lastly, a final concentration of 300 mM NaCl, 50 mM HEPES pH 7.5 was added to the mixture and incubated at 23 °C for 24 h to assemble co-assembled capsids. Co-assembled capsids were then purified with AKTApurier FPLC. To remove hexamers from co-assembled capsids and create holey capsids, purified capsids were mixed with 3 M urea and 100 μM EDTA, and incubated at 23 °C for 24 h. Holey capsids were further purified using an Amicon Ultra-Cel with 100 kDa cutoff to remove disassembled Cp149$_{His}$Cp149$_{Y132A}$ subunits with 150 mM NaCl, 50 mM HEPES pH 7.5 as washing buffer and repeated four times. Cp150 was labeled by BODIPY-FL as published previously[39]. BODIPY-FL-labeled Cp150, Cp150Bo (2 μM) was mixed with 4 μM holey capsid first and refilling of the holey capsids was initiated by increasing NaCl concentration to 300 mM and incubating at 23 °C for 24 h. Cp150Bo (2 μM) was mixed with 300 mM NaCl as a control for the refilling reaction. To detect refilling of the holey capsids, absorbance at 280 nm for protein and 504 nm for BODIPY of each sample was collected on HPLC with 4 nm bandwidth. Fluorescence of Cp150Bo at 512 nm was collected with excitation wavelength at 504 nm using a Shimazu RF-20A detector with a 20 nm spectral bandwidth.

**Charge detection mass spectrometry**. CDMS allows the mass distribution to be measured for heterogeneous and high-molecular-weight samples that are not accessible by conventional MS. In CDMS, the masses of individual ions are determined from simultaneous measurements of each ions mass to charge ratio (m/z) and charge (z). Mass measurements are performed for thousands of ions and then the results are binned to yield a mass distribution. The homebuilt second-generation CDMS instrument employed here has been described previously[67–69]. Samples were electrosprayed using a commercial nanoelectrospray source (Advion Triversa Nanomate®) and the resulting ions enter the instrument though a heated metal capillary. The ions pass through several stages of differential pumping to separate them from the ambient gas. They are then accelerated to a nominal energy of 130 eV/z and focused into a dual hemispherical deflection energy analyzer (HDA), which transmits a narrow band of ion energies centered on the nominal ion energy. Ions that pass through the HDA are focused into an electrostatic linear ion trap where trapped ions oscillate back and forth through a detection cylinder, in this case for 100 ms). When an ion enters the cylinder, it induces a charge that is detected by a charge-sensitive preamplifier[70]. The resulting signal is amplified, digitized, and analyzed using a fast Fourier transform (FFT)[71]. The oscillation frequency is related to the ion's m/z and the FFT magnitudes are proportional to the charge. Prior to electrospray it is necessary to buffer exchange samples into a volatile buffer. In this case, hexamer samples were buffer exchanged into 20 mM ammonium acetate using SEC (Micro Bio-Spin P-6 Gel Columns, BIO-RAD).

**Resistive-pulse sensing**. Nanofluidic devices for RPS measurements were fabricated from D263 glass substrates. Each device has two V-shaped microchannels that are connected by a series of nanochannels and nanopores[54,72] (Supplementary

Fig. 7a). Nanochannels and nanopores (Supplementary Fig. 7a) were milled into the 10 μm gap between the V-shaped microchannels with an Auriga 60 focused ion beam instrument (Carl Zeiss AG). The substrate was then bonded to a D263 cover glass at 545 °C for 12 h. Reservoirs were then attached to the RPS device with epoxy.

Multicycle resistive-pulse sensing[54] was used to measure Cp150 capsid standards, hexamer nucleated capsids, holey capsids, and surface refilled capsids. For these measurements, capsid samples were diluted with HEPES buffered 1 M NaCl to suitable concentrations (~0.1 μM dimer). For ping-pong experiments, we used a device with four pores in series. A potential was applied across the nanochannel and current was monitored with an Axopatch 200B (Molecular Devices, LLC) connected through a data acquisition card (PCIe-6353, National Instruments). Changes in current were analyzed, in real time, with a LabVIEW program (National Instruments). A threshold for pulses was determined by comparison to the baseline current. To accomplish a ping-pong experiment, after a pulse was detected for each of the four nanopores, the potential was reversed. For each particle used in the reported size distributions, we recorded at least 84 pulses, equivalent to 10.5 cycles back and forth through the RPS device. The two most abundant populations shown in the size distributions corresponded to $T = 3$ capsids and $T = 4$ capsids, and their peak positions were aligned to their corresponding numbers of dimers (90-mer and 120-mer). For easier visualization of populations relative to each other, the histograms of capsids in each sample were normalized to the same total number of counts.

**Negative-stain EM and data processing**. $Cp149_{His}Cp149_{Y132A}$ hexamers were purified using AKTApurier FPLC and 4 μL of such sample was applied to a glow-discharged carbon film with a 300-mesh Cu grid immediately and stained with 0.75% uranyl formate. During the grid preparation, the grid was washed twice using $ddH_2O$, blotted using filter paper, and air dried. Images were collected on a JEOL JEM 1400Plus transmission microscope equipped with a Gatan Oneview 4k × 4k COMS camera at a nominal magnification of ×50,000 using the low-dose conditions. A total of 165 micrographs was collected and 55,261 particles were semi-auto-boxed using e2boxer_old.py from EMAN2[73] and subjected to two rounds of reference-free 2D classification using RELION (version 2.1)[74]. Selected 900 particles from 6 classes with clear definition for hexamers were used to build an initial model with C1 symmetry. Overall, 5755 particles were selected for hexamer 3D auto-refinement with C6 symmetry, resulting in a 17 Å hexamer density map (Supplementary Table 1).

**Cryo-EM and holey capsid reconstruction**. Samples of holey capsids were purified through the AKTApurier FPLC and concentrated to 20 μM using an Amicon Ultra-Cel with 100 kDa cutoff. Grids preparation, data collection, and image processing followed previously published protocol[33]. Simply, 4 μL of the concentrated holey capsids was applied to a glow-discharged UltraAuFoil R2/2 holey gold film grid and repeated four times to enrich particles on the grid[75] and then vitrified using a ThermoFisher Vitrobot (Mark IV, blot time: 15 s, blot force: 0, wait time: 25 s). Grids were imaged using a Thermo Scientific Talos Artica microscope operated at 200 kV. Low-dose (~30 $e^{-1}$ Å$^2$) cryo-EM images, at a nominal magnification of ×120,000 (1.128 Å per pixel), were collected on a Falcon III direct electron detecting camera in electron counting mode with defocus range from −1 to −3 μm. A total of 1366 movie frames was collected using the EPU automated data collection software. Contrast transfer function parameters were estimated using ctffind4[76].

Total of 13,904 particles were manually boxed out of 300 micrographs using the e2boxer_old.py from EMAN2[73] and subjected to reference-free 2D classification using RELION (version 2.1)[74]. The 2D classes with clear density were selected as reference templates for particle auto-picking from all micrographs using Auto-Picking software in RELION. A total of 65,728 particles were auto picked and subjected to reference-free 2D classification in RELION[74]. After discarding classes with blurred density and deformed structures, 41,057 particles in 21 classes were left for further reconstruction. The same 21 classes were used to build an initial model de novo using C1 symmetry in RELION. To capture all possible capsid conformations, C1 symmetry was applied through all 3D refinement process. The initial model was used as template for 3D classification to separate class of complete capsids and class of holey capsids. Particles (23,478) from the closed shell capsid class was auto-refined to reconstruct a complete capsid model. The rest 17,579 particles were further divided into two classes, 9697 particles for class with a small hole and 7882 particles for another class with a big hole, in a 3D classification process. All three classes were refined independently using their own models from the 3D classification results, a complete capsid, a holey capsid with a small hole, and a holey capsid with a big hole. Only the complete capsid model had enough particles to reach a sub-nanometer resolution for B-factor sharpening in post-processing step. Lastly, the complete capsid was refined to 6.2 Å with a calculated B-factor of −203 Å$^2$, the holey capsid with small hole was refined to 10.8 Å and the holey capsid with a big hole was refined to 12.9 Å (Supplementary Table 2).

## Data availability
The data that support the findings of this study are available from the corresponding author upon reasonable request. Electron Microscopy Data Bank (EMDB) accession code for full capsid is EMD-22132, for a Ni-induced hexamer of dimers is EMD-22133, for a holey capsid with a single hexamer-sized hole is EMD-22134, and for a holey capsid with a double hexamer hole is EMD-22135. Details describing these reconstructions are in Supplementary Tables 1 and 2.

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

## Acknowledgements

We gratefully acknowledge support for this study from the NIH by R01 AI118933 to A.Z. and R01 GM129354 to S.C.J. We gratefully acknowledge the use of the Laboratory for Biological Mass Spectrometry at Indiana University (IU) for the data in Fig. 2c, the IU Electron Microscopy Center, and IU Nanoscale Characterization Facility. Electron microscopy at the IU Electron Microscopy Center was partially supported by a CTSI Spring 2019 grant to A.Z.

## Author contributions

Z.Z. designed and conducted the overall study, including protein purification, in vitro solution characterization, and sample preparation for Cryo-EM, RPS, and CDMS characterization. A.Z. contributed to the initiation and design of the study. J.C.W. collected Cryo-EM images and helped Z.Z. to perform reconstruction. M.Z. performed the RPS experiment and analyzed the RPS data. S.C.J. directed the RPS experiment and data analysis. N.A.L. performed the CDMS experiment and analyzed the CDMS data. M.F.J. directed the CDMS experiment and data analysis. This paper was mainly prepared by Z.Z. and A.Z. J.C.W., M.F.J., and S.C.J. helped to edit the manuscript.

## Competing interests

The authors declare no competing interests.

**Additional information**

