## [Peer Review File · Nature Communications]

REVIEWER COMMENTS

Reviewer #1 (Remarks to the Author):

In this manuscript, the authors report the preparation of asymmetric hepatitis B virus capsids by hierarchical assembly of engineered coat proteins. To this end, the native 149-residue coat protein (CP149) was engineered and modified, resulting in a heterodimer (CP149hisCP149Y132A) that displays a programmable binding domain (CP149his) and a conditional stop (CP149Y132A). In presence of Ni²⁺, the heterodimer assembly into hexamers that subsequently nucleate the capsid formation in presence of cysteine-modified coat protein (CP150). Under EDTA and urea conditions, the heterogeneous capsid shows partial disassembly, which results in holey structures. Finally, the authors reverse the hole in the capsid with fluorescent-labeled coat proteins (CP150Bo).

Overall, the manuscript is interesting for the specific field but lays short in reaching the wider community. Additionally, the authors fail to prove convincingly several statements in the manuscript, and extensively experimental work would be required to support their hypotheses, as detailed below. Therefore, I recommend not publishing this manuscript in Nature Communications, due to not reaching the high standards of this journal.

The main claim of the paper, which is the achievement of a controlled, hierarchical assembly of a protein cage is interesting for the specialized virology/physical virology field. However, the research questions reaching out the wider field such as external and internal modification of the capsid, while outlined in the abstract (line 39), are not addressed in the manuscript.

Regarding the experimental aspect of the paper, the major concern regards the holey capsid refilling section (line 267-288). The holey capsids are refilled with fluorescently-labeled CP150Bo, obtained by attaching a fluorophore to the cysteine 150. The refilled capsids are characterized by resistive pulsed sensing (RPS), showing a decrease in the intermediate particles. Size exclusion chromatography (SEC, Fig. 4c) shows, on the other hand, no solid proof of the re-assembly. First, the emission results are not conclusive when referring to population without the associated absorption chromatograms at 280 nm (protein content) and 504 (BODIPY absorption). Only if the protein concentration is similar, the increase in fluorescence is meaningful. Second, the authors fail to prove that the fluorescent signal eluting at c.a. 8 mL (Figure 4c, blue) is, in fact, refilled capsids, instead of CP150Bo homogeneous capsids co-eluting with CP150 holey capsids. The justification of fluorescence quenching in line 282-283 is not enough by itself, as a decreased signal might be due to a low protein concentration, hence the need of the abovementioned chromatogram showing the absorption at 280 nm. Experimental details such as excitation wavelength and bandgaps employed are missing.

Other concerns are the following:

The design and preparation of heterodimers (CP149hisCP149Y132A) is well described and their assembly into hexamers is thoroughly characterized by SEC, charge detection mass spectrometry, SDS-PAGE and negative-stain EM reconstruction. A suitable control experiment is included in Supplementary Fig. 2. Minor concerns arise, such the need of a bigger font size in Figure 2a and the inclusion of low magnification, raw negative-stain EM pictures of the hexamers and double hexamers in the SI.

CP149hisCP149Y132A hexamers co-assemble with cysteine-bearing CP150 homodimers. The authors justify the selection of this homodimer based on the crosslinking ability of the heterodimers and subsequent capsid sturdiness upon the nucleating patch removal (line 188-189). Further evidence is necessary to support the importance of this choice: is the assembly of the homogeneous and heterogeneous CP150 capsids reversible in presence of oxidizing agents, proving the key role of the disulfide bond in the overall stability? Is native CP149 able to form heterogeneous capsids? Is the resulting capsid not strong enough to undergo the nucleating patch removal?

In order to assemble the heterogeneous capsids, a CP149hisCP149Y132A/ CP150 ratio of 4:1 is selected (line 77-78). A statement or experimental evidence justifying this selection will be highly appreciated.

The authors state that the CP150 led to "morphologically normal capsids" in presence of CP149hisCP149Y132A heterodimer with no evidence of subunit segregation (line 196-198). However, proof of this statement is based exclusively in SEC (Supplementary Fig. 3) and further experimental data would be necessary to support this statement (cryo-EM at the very least). Additionally, supplementary Fig. 3 will benefit from a dimer-to-capsid area ratio if the dimer-to-capsid population is discussed between different mixtures (Supplementary Fig. 3, caption).

The resection of the nucleating patch was reportedly achieved and characterized by cryo-EM reconstruction and RPS. The latest technique shows a constant population of $T = 3$ particles and a decrease in the $T = 4$ population in the holey particles, with the appearance of intermediate species (Fig. 4b). This is the first time that two different capsid morphologies are addressed in the introduction or results and discussion sections. How this heterogeneous population affect to the cryo-EM reconstruction process? A deeper background description is needed for researchers unfamiliar with the hepatitis B virus capsid. Controls such as the CP150 capsid are claimed to be used as standards (line 96), but no evidence is shown. Is the $T = 3/ T = 4$ ratio the same in the homogeneous and the heterogeneous particles? Does the CP149hisCP149Y132A/ CP150 ratio employed for the assembly affect the population? Control experiments with heterogeneous CP149hisCP149Y132A-CP149 capsid will enrich the discussion.

Additionally, few typos were found over the text (e.g. line 208, 233, 283...), reference format (e.g. ref 58, line 457), and missing reference to figure panels in the text (e.g. Fig. 3a). Therefore, an exhaustive revision of the manuscript overall format is strongly suggested.

Reviewer #2 (Remarks to the Author):

This paper describes experiments to exploit the hierarchical nature of HBV capsid assembly to create asymmetric assemblies, in this case structures where the underlying icosahedral symmetry is broken by the presence of a small number of mutated subunits. Designing and controlling self-assembling protein architectures is an area of great current interest. Several recent successes have been demonstrated in creating novel cages by a few different approaches. Here, a natural capsid, HBV, is chosen for a target of de-symmetrization. This is an interesting twist on current developments, as breaking symmetry provides potential advantages in control and subunit addressing. The current study involves several clever strategic elements that lead to success in this challenging task. The experimental work appears solid, and it's a strength that several different methods are brought to bear on characterization. There is some room for improving clarity in some spots in the manuscript and in phrasing a few things a bit more carefully, as detailed below, but otherwise the paper provides a strong description of leading-edge protein design work and represents an important addition to the field.

1) More care is needed in lines in describing the dimer assembly. On line 58, does 'adjacent subunit' mean from a different dimer? On line 60, "for which each monomer can assemble", does this mean with monomers from adjacent dimers?

2) The hierarchical nature of HBV assembly guides things toward the desired outcomes (a dedicate patch of distinct character, less tightly bound than the rest of the capsid), but not much is said of how, and how strictly, other outcomes are prevented. Evidently, growth tends to expand from an initial hexamer of dimers, but can another copy of that hexamer not be accidentally added in some lower yield, leading to capsids with two special patches?

- 3) The phrasing in the paper of a capsid as a Janus particle with two patches leads to confusion early; it only becomes clear later that the capsid is really mainly uniform, except for one small patch (6 special dimers out of 120 dimers total). This needs to be rephrased and clarified earlier.
- 4) Figure 1 doesn't do as much as it could to explain the clever multi-step procedure. The cross-linking element that allows subsequent removal of the special patch later isn't included.
- 5) Line 136-137 needs to be rephrased, as it currently indicates that the gene construct contained "two genes for each monomer".
- 6) It is surprising and notable, and critical to success of the present design, that nearly exclusive formation of a heterodimer is obtained from bi-cistronic expression. What prevents homodimer formation? There doesn't appear to be any energetic/thermodynamic contribution, so does this have something to do with the tendency of the two chains coming sequentially off the same ribosome to co-assemble? What do the authors suggest is going on here. It is very unusual.
- 7) Should CDMS be described as a variation on a native mass spec method to help the reader understand that the masses of intact, non-covalent assemblies are being measured?
- 8) The paper talks in several places about the capsid getting "refilled" after removal of the special patch. This needs to be rethought, since nothing is actually going inside the capsid. Maybe the shell is being re-annealed, or re-assembled, or re-sealed, or repaired? Or maybe the *surface* of the shell is being refilled.
- 9) The authors need to be a bit more circumspect in how they talk about the generalizability of what has been done here. The present approach relies on a system with *multiple levels* of hierarchical assembly control; the dimer must be preparable separate from the hexamer of dimers, and likewise for the hexamer of dimers. And a rather narrow thermodynamic window has to be hit by mutagenesis in order to get the whole system to work. It's a testament to the difficulty of what was accomplished here, and the unlikeliness (in this reviewer's opinion) of extending it to very different systems doesn't take away from the current success. The authors should be more specific about what would be required to extend the ideas to other systems.
- 10) On a similar point, the authors indicate that "not every viral capsid is built from homodimers". That's an understatement! It's rather rare, so this needs to be rephrased, and if there are other candidate capsids based on similar hierarchical assembly of dimers then that more specific information would be useful to readers.
- 11) In describing prior work on engineering protein capsids and cages, the authors should consider rephrasing in line 43 where they cite what other authors have "attempted". Given the successes that have been achieved by the cited papers in this area, a different word choice might make sense: "...have exploited the advantages of natural systems..."?

Reviewer #3 (Remarks to the Author):

Zlotnick and colleagues describe a hierarchical assembly of asymmetric HBV capsids and use an impressive array of analytical techniques (cryoEM, CD-MS and RPS) to characterize the products.

The cp149-Y132A constructs and its co-assembly with 'WT' cp149 has been described before in refs 34-36. Here, novel constructs for formation of a cp149//cp149-Y132A heterodimer is described. In addition, a procedure to assemble a patchy particle, using hexamers and double hexamer seeds composed of the heterodimer, is described. It is shown that this heterodimer patch can be selectively removed and subsequently filled out with additional building blocks.

In the abstract of the manuscript, the authors claim that 'this strategy can be generalized', but no explanation or demonstration of this bold claim is given. It seems this strategy is actually rather specific to HBV, and only applicable to capsids with dimeric building blocks, with a nucleation-elongation type assembly pathway.

In the introduction the authors state that a current shortcoming in the field of virus-based nanoplatfoms is that asymmetric particles cannot yet be assembled, limiting applications. A discussion is missing about which particular applications would require such features.

It is not yet entirely clear to me if the procedure yields 'pure' composite particles, or also contains a substantial fraction of full (wt cp149) homodimer particles. On that note, the complete absence of any homodimers in figure 2c is remarkable and deserved some additional discussion/explanation.

Another point that is not clear is how the T=3 particle is formed in the hierarchical assembly procedure as the manuscript almost entirely focusses on the T=4 structure. The RPS data in figure 4b cuts out the region below T=3 that might show any defected particles. Could these same distributions also no be assessed by CD-MS for defects in both T=3 and T4 particles? In any case, the RPS data suggests that removal of heterodimers from assembled particles results in much more heterogeneous defects than only hexamers/double hexamers.

On p9 line 153, it is written that the constructs could tolerate larger insertions for antigen presentation, but no such data is shown.

On p10 line 172, the resolution estimate of the negative stain model is in my opinion a bit inflated, and reported with too much precision (is it 17 or 18 angstrom? or actually just about 2 nm?).

On p10 line 178, it is written that different sizes of assembly seeds can be formed with other mutations, but no evidence of this is given, nor any explanation.

On p13/14, the discussion of the size of the hole in the cryoEM reconstructions is over interpreting the densities in my opinion. I don't think that hexamer or double hexamer defects are effectively sorted in the image classification, or that individual images are 100% accurately aligned, and that therefore the apparent size of the hole in the map is a convoluted effect of these image processing artefacts, sample heterogeneity and the actual physical size of the hole in the capsids.

In figure 3b, the mass of the assigned double hexamer is wrongly annotated and I also believe that the assignment of that signal as being a double hexamer is somewhat wishful. What would the theoretical mass of the annotated complexes be and how accurate are the experimental masses?

In Figure 4c, the lower fluorescence signal of the cp150Bo capsids is interpreted as a fluorescence quenching effect. Can the authors exclude that no quenching takes place, but that the fluorescence signal of the assembled capsid is simply lost against the high background of free dimers?

Author's Response to Reviewer #1:

In this manuscript, the authors report the preparation of asymmetric hepatitis B virus capsids by hierarchical assembly of engineered coat proteins. To this end, the native 149-residue coat protein (CP149) was engineered and modified, resulting in a heterodimer (CP149hisCP149Y132A) that displays a programmable binding domain (CP149his) and a conditional stop (CP149Y132A). In presence of Ni²⁺, the heterodimer assembly into hexamers that subsequently nucleate the capsid formation in presence of cysteine-modified coat protein (CP150). Under EDTA and urea conditions, the heterogeneous capsid shows partial disassembly, which results in holey structures. Finally, the authors reverse the hole in the capsid with fluorescent-labeled coat proteins (CP150Bo).

1) Overall, the manuscript is interesting for the specific field but lays short in reaching the wider community. Additionally, the authors fail to prove convincingly several statements in the manuscript, and extensively experimental work would be required to support their hypotheses, as detailed below. Therefore, I recommend not publishing this manuscript in Nature Communications, due to not reaching the high standards of this journal.

Response:

With this revision, we address the reviewer's concerns on technical points including demonstration of refilling the surface of holey capsids, purification of heterodimer, and the rationale for using the Cp150 construct for growth on the heterodimer nucleus. We also explain the basis for proposing future directions for hierarchical system developed in this paper.

2) The main claim of the paper, which is the achievement of a controlled, hierarchical assembly of a protein cage is interesting for the specialized virology/physical virology field. However, the research questions reaching out the wider field such as external and internal modification of the capsid, while outlined in the abstract (line 39), are not addressed in the manuscript.

Response:

Reviewers 1 and 2 both indicate that we need to explicitly show that our work has general impact. Most of this deficiency is now addressed by incorporating explanations of the relationship of our work to Janus particles and the importance of being able to impose asymmetry and hierarchical assembly. These discussions at the end of the introduction (line 71-72) and in the conclusion (line 343-353) provide context for our study. As noted by reviewer 1, it would be imposing to experimentally demonstrate these applications. However, the goal of our work is to demonstrate the platform. The lab envisions a series of subsequent papers to provide specific examples.

Notably, in the first paragraph of the conclusions we identify where modifications can be added to the system. We do not go into great detail as these are not experiments performed in the paper. These are opportunities for future investigation.

As a second paragraph of conclusions we provide a strategy for modifying self-assembling proteins and developing hierarchical assembly. This strategy can be obviously applied to viruses but can also work with any biological or abiological self-assembling system. The emphasis of the paragraph is creating attainable thermodynamic boundaries between steps. This is further discussed in our response to reviewer 2.

3) Regarding the experimental aspect of the paper, the major concern regards the holey capsid refilling section (line 267-288). The holey capsids are refilled with fluorescently-labeled CP150Bo, obtained by attaching a fluorophore to the cysteine 150. The refilled capsids are characterized by resistive pulsive sensing (RPS), showing a decrease in the intermediate particles. Size exclusion chromatography (SEC, Fig. 4c) shows, on the other hand, no solid proof of the re-assembly. First, the emission results are not conclusive when referring to population without the associated absorption chromatograms at 280 nm (protein content) and 504 (BODIPY absorption). Only if the protein concentration is similar, the increase in fluorescence is meaningful. Second, the authors fail to prove that the fluorescent signal eluting at c.a. 8 mL (Figure 4c, blue) is, in fact, refilled capsids, instead of CP150Bo homogeneous capsids co-eluting with CP150 holey capsids. The justification of fluorescence quenching in line 282-283 is not enough by itself, as a decreased signal might be due to a low protein concentration, hence the need of the abovementioned chromatogram showing the absorption at 280 nm. Experimental details such as excitation wavelength and bandgaps employed are missing.

Response:

We thank the reviewer for pointing this out. In addition to supplementary data, we show that Cp150Bo is associated with capsids and is far more fluorescent than for a capsid made exclusively of Cp150Bo. In an analysis of these data we find that for about 13% of the protein eluting with refilled holey capsids is Cp150Bo. On average, 13% corresponds to about 16 Cp150Bo dimers per T=4 capsid, in agreement with size distributions determined by RPS and by cryo-EM.

In supplementary information, we have added absorbance chromatographs at 280 nm and 504 nm (Supplementary Fig. 10), collected simultaneously with the samples in Fig. 4c that were detected by fluorescence. The chromatographs are for 4 μ M Holey capsids, 4 μ M Holey capsids + 2 μ M Cp150Bo, and 2 μ M Cp150 assembly. Experimental details for absorbance and fluorescence data detection have been added to supplementary information (line 112-115).

The following discussion is also added to supplementary information (starting at line 266) with a summary of it to our draft (line 296-302):

For the holey capsid sample, we did not observe BODIPY absorbance at 504 nm or fluorescence (Supplementary Fig. 10, and Fig. 4c). For the Cp150Bo assembly control, we observed absorbance at 280 nm and 504 nm for both capsids and dimers. The assembled capsids and the unassembled dimers roughly have the same peak areas in both absorbance chromatographs, indicating they have the same amount of protein. However, we observed little BODIPY fluorescence for capsids compared to the strong fluorescence of unassembled dimers (Fig 4c), which is consistent with the fluorescence quenching that we previously reported for these capsids ¹. However, for the refilled holey capsid sample we observed substantial fluorescence resulting from unquenched fluorophores. As noted in the paper, we did not observe free dimer, indicating they had assembled or were bound into relative high affinity sites.

	Cp150Bo at 7.5mL (mAU)	Refilled capsid at 7.5mL (mAU)
Abs280	1.1	7.4
Abs504	1.9	1.7
Abs280/ Abs504 for Cp150Bo	0.579	Not relevant
A280 attributed to Cp150Bo	1.1	0.98

These absorbance data let us quantify the amount of Cp150Bo in the capsid peak. The data were collected on an HPLC equipped with a diode array detector and a fluorescence detector, so that the whole spectrum is measured at one time. Using the A280/A504 for the Cp150Bo capsid peak, we determine an absorbance normalization $n_{A280/A50}$. By multiplying the A504 for the refilled capsid peak by n we obtain the amount of A280 absorbance related to Cp150Bo. Thus, the fraction of Cp150Bo ($X_{Cp150Bo,capsid}$) in the refilled capsid peak is

$$\begin{aligned}
 X_{Cp150Bo,capsid} &= (A504_{refilled} \times n_{A280/A50}) / A280_{refilled} \\
 &= (1.7 \times 0.579) / 7.4 \\
 &= 0.133
 \end{aligned}$$

We found that Cp150Bo accounts for ~13% of the absorbance of the refilled capsid peak. If all of these capsids were T=4 particles, this gives an average of ~16 Cp150Bo dimers per capsid. It agrees well with our RPS data and Cryo-EM data, showing a hole missing ~9 to 18 dimers. We cannot exclude the presence of some homogeneous Cp150Bo capsids and some T=3 particles in the refilled holey capsid sample. The loss of the intermediate peak in the RPS of the refilled sample and the loss of free Cp150Bo dimer in the chromatogram, argue that we have found a new repository for Cp150Bo. Therefore, we surmise that the majority of the BODIPY absorbance arose from the refilled Cp150Bo dimers.

Other concerns are the following:

4) The design and preparation of heterodimers (CP149hisCP149Y132A) is well described and their assembly into hexamers is thoroughly characterized by SEC, charge detection mass spectrometry, SDS-PAGE and negative-stain EM reconstruction. A suitable control experiment is included in Supplementary Fig. 2. Minor concerns arise, such as the need for a bigger font size in Figure 2a and the inclusion of low magnification, raw negative-stain EM pictures of the hexamers and double hexamers in the SI.

Response:

Done.

We have increased the font size in all panels of Fig. 2.

Also, we added a negative stain micrograph of a hexamer sample as Supplementary Fig. 3 referenced at line 182 of the manuscript. In this image, hexamers are highlighted by red arrows and double hexamers by blue arrows.

5) CP149hisCP149Y132A hexamers co-assemble with cysteine-bearing CP150 homodimers. The authors justify the selection of this homodimer based on the crosslinking ability of the heterodimers and subsequent capsid sturdiness upon the nucleating patch removal (line 188-189). Further evidence is necessary to support the importance of this choice: is the assembly of the homogeneous and heterogeneous CP150 capsids reversible in presence of oxidizing agents, proving the key role of the disulfide bond in the overall stability? Is native CP149 able to form heterogeneous capsids? Is the resulting capsid not strong enough to undergo the nucleating patch removal?

Response:

The choice of Cp150 in this study was to provide a subunit whose assembly could be nucleated by hexamer, and then could be switched into a form that was refractory to dissociation, allowing us to remove hexamer. In Cp150, the 3 cysteines of the native assembly domain (Cp149) are mutated to alanine and an additional cysteine is added to the C-terminus. Cp150 capsid oxidize spontaneously in air, we do not introduce oxidizing agents. The exceptional stability of Cp150 capsids has facilitated crystallographic and cryo-EM studies of HBV capsids with antiviral compounds that distort inter-subunit geometry² as Cp149 capsids become too irregular for structure determination.

Reduced Cp150 assembles and disassembles readily. Both Cp149 and Cp150 dimers undergo essentially the same purification scheme starting with capsids isolated from *E. coli*. During purification, excessive DTT is added to minimize and reverse oxidation.

Capsids are quantitatively disassembled into dimers by 3 M Urea, and the dimers purified by size exclusion chromatography.

Oxidized Cp150 capsids are stable enough to withstand hexamer removal by 3M urea, though reduced Cp150 and Cp149 capsids are not. In a previous effort to develop patchy capsids, we mixed Cp150 with either Cp149 or Cp150 where the C-terminal cysteine was passivated by N-ethyl maleimide or BoDIPY. The ratio of Cp150 to passivated subunit was varied. We found that the majority of passivated subunit could be removed with up to 5M urea and the complexes assayed by RPS. Though presumably there was a distribution of sizes, capsids where less than 25% of the subunits had been removed could be isolated and refilled. In these studies, we found that for random incorporation of defects, 25% represented a percolation limit beyond which capsids fragmented. This result was confirmed by Monte Carlo simulations.³

The bottom line is that reduced Cp150 behaves nearly the same as Cp149 while oxidized Cp150 forms an extremely stable complex.

We added a new sentence at line 200-201 to explain the stability of Cp150 capsids.

6) In order to assembly the heterogeneous capsids, a CP149hisCP149Y132A/ CP150 ratio of 4:1 is selected (line 77-78). A statement or experimental evidence justifying this selection will be highly appreciated.

Response:

In fact, we did try different ratios. We found that with 4:1 and 10:1 Cp150:heterodimer ratios there was no significant differences in the amount of heterodimer that eluted in the capsid fraction. The reason we selected the 4:1 ratio is that we wanted to enrich the relative concentration of hexamer nuclei and minimize Cp150 dimer self-assembly.

Now we have modified the sentence at line 85-86.

7) The authors state that the CP150 led to “morphologically normal capsids” in presence of CP149hisCP149Y132A heterodimer with no evidence of subunit segregation (line 196-198). However, proof of this statement is based exclusively in SEC (Supplementary Fig. 3) and further experimental data would be necessary to support this statement (cryo-EM at the very least).

Response:

We showed that capsids were morphologically normal based on SEC (Fig 4a) and negative stain EM (new Supplementary Fig. 5). It would be very difficult to isolate heterodimers in cryo-EM micrographs and reconstructions if they were not conspicuously labeled.

To show that hybrid dimers do not segregate when co-assembled with Cp 150, we compared disassembly of capsids where heterodimer and Cp150 dimers were co-assembled to capsids where heterodimer hexamers nucleated Cp150 assembly (below). In this experiment, we treated both types of capsids with EDTA and urea to remove clusters of heterodimers; capsids were isolated and evaluated with SDS-PAGE. We did not observe obvious loss of the heterodimer in the co-assembled capsids, suggesting heterodimers were generally surround by crosslinked Cp150. Whereas in the hexamer + Cp150 capsids, heterodimers basically disappeared, indicating they are labile and thus probably in patches with other heterodimers. If there were segregated regions of heterodimer in the heterodimer + Cp150 capsids, we would expect a similar result. Below is the SDS-PAGE of our experiment.

8) Additionally, supplementary Fig. 3 will benefit from a dimer-to-capsid area ratio if the dimer-to-capsid population is discussed between different mixtures (Supplementary Fig. 3, caption).

Response:

Because we think we have already condensed a broad range of information in this manuscript to introduce our new assembly system, we decided not to perform a detailed comparison of assembly conditions this time. However, we really appreciate this idea. This is a component of our future plans to investigate how to produce more homogeneous holey capsids.

9) The resection of the nucleating patch was reportedly achieved and characterized by cryo-EM reconstruction and RPS. The latest technique shows a constant population of

T = 3 particles and a decrease in the T = 4 population in the holey particles, with the appearance of intermediate species (Fig. 4b). This is the first time that two different capsid morphologies are addressed in the introduction or results and discussion sections. How this heterogeneous population affect to the cryo-EM reconstruction process? A deeper background description is needed for researchers unfamiliar with the hepatitis B virus capsid. Controls such as the CP150 capsid are claimed to be used as standards (line 96), but no evidence is shown. Is the T = 3/ T = 4 ratio the same in the homogeneous and the heterogeneous particles? Does the CP149hisCP149Y132A/CP150 ratio employed for the assembly affect the population? Control experiments with heterogeneous CP149hisCP149Y132A-CP149 capsid will enrich the discussion.

Response:

We have added information about the two types of HBV capsids, T=3 and T=4, in the introduction at line 60-62. T=3 particles are found in natural infections, expression systems, and assembly of purified protein. It is usually a minor component ca. 10%. The T=3 particles are easily differentiated in reconstructions due to the difference in diameter readily seen in micrographs (~30nm vs ~36nm). In reconstructions, T=3 particles are easily separated during 2D classification. For biochemical use, as standards, T=3 and T=4 particles are separated by sucrose gradient. Because we did not observe obvious breaks or holes on T=3 capsids in cryo-EM, we did not include a reconstruction of T=3 capsids in the paper.

We now include (Supplementary Fig. 9) an example of Cp150 T=3 and T=4 particles as standards for RPS. We also added sentence at line 234 to refer to the new figure.

We did not conduct experiments to determine the T=3/T=4 ratio for heterodimer assembly. As can be seen in figure 4c, during hexamer-nucleated assembly T=3 particles are only about 25% of the capsids but we do not know whether they include heterodimer. As we did not observe holey T=3 capsids, we suspect that most of the T=3 capsids were not nucleated by hexamers and are thus pure Cp150 homodimer. We feel that a detailed analysis of the T=3 particles will distract from the focus on redesigned assembly paths. This point is now explicitly stated on lines 239-240.

10) Additionally, few typos were found over the text (e.g. line 208, 233, 283...), reference format (e.g. ref 58, line 457), and missing reference to figure panels in the text (e.g. Fig. 3a). Therefore, an exhaustive revision of the manuscript overall format is strongly suggested.

Response:

We have addressed the reviewers' observations and have carefully proofread.

Author's Response to Reviewer #2:

This paper describes experiments to exploit the hierarchical nature of HBV capsid assembly to create asymmetric assemblies, in this case structures where the underlying icosahedral symmetry is broken by the presence of a small number of mutated subunits. Designing and controlling self-assembling protein architectures is an area of great current interest. Several recent successes have been demonstrated in creating novel cages by a few different approaches. Here, a natural capsid, HBV, is chosen for a target of de-symmetrization. This is an interesting twist on current developments, as breaking symmetry provides potential advantages in control and subunit addressing. The current study involves several clever strategic elements that lead to success in this challenging task. The experimental work appears solid, and it's a strength that several different methods are brought to bear on characterization. There is some room for improving clarity in some spots in the manuscript and in phrasing a few things a bit more carefully, as detailed below, but otherwise the paper provides a strong description of leading-edge protein design work and represents an important addition to the field.

1) More care is needed in lines in describing the dimer assembly. On line 58, does 'adjacent subunit' mean from a different dimer? On line 60, "for which each monomer can assemble", does this mean with monomers from adjacent dimers?

Response:

We appreciate the reviewer for pointing this out. We have changed the description of the assembly process in our updated manuscript at line 63-64.

2) The hierarchical nature of HBV assembly guides things toward the desired outcomes (a dedicate patch of distinct character, less tightly bound than the rest of the capsid), but not much is said of how, and how strictly, other outcomes are prevented. Evidently, growth tends to expand from an initial hexamer of dimers, but can another copy of that hexamer not be accidentally added in some lower yield, leading to capsids with two special patches?

Response:

It is a very interesting question. In our cryo-EM micrographs of holey capsid samples, we did observe a few capsids with two breaks, potentially two holes. But they are very rare. There are three major explanations for the relative rarity of capsids with two independent holes. First, as HBV capsid assembly in 300mM NaCl is a fast and downhill reaction and hexamers are effective nuclei, the hexamers may rapidly surround themselves with free dimer; these progressively larger intermediates are less able to join together. Second, in our experimental setup, we have hexamers enough to nucleate assembly but may not have enough to have multiple hexamers in a single capsid. By manipulating the ratio of hexamers to Cp150 dimers we may be able to acquire different results, e.g. two or three holes on one holey capsid. A third possibility is that hexamers may have sub-optimal geometry for a complete capsid and are bounded by the Y132A mutant; incorporation of two or more hexamers into a nascent capsid may result in a

complex that is unstable (prior to disulfide formation) and dissociates.

3) The phrasing in the paper of a capsid as a Janus particle with two patches leads to confusion early; it only becomes clear later that the capsid is really mainly uniform, except for one small patch (6 special dimers out of 120 dimers total). This needs to be rephrased and clarified earlier.

Response:

The reviewer is correct. We were excited by the ability to create a particle with two surface domains (not two patches in a background). Though the original design indicated that the smaller domain was precisely 6 dimers, we observed experimentally that most holes are in the range of 9-15 dimers – still a relatively small patch. We have revised the abstract and introduction to clarify the distinction between our construct and an archetypal Janus particle. The advantages of Janus-like particles are described in the final paragraph of the conclusion.

4) Figure 1 doesn't do as much as it could to explain the clever multi-step procedure. The cross-linking element that allows subsequent removal of the special patch later isn't included.

Response:

We have modified our Fig. 1 to explicitly note crosslinking step.

5) Line 136-137 needs to be rephrased, as it currently indicates that the gene construct contained “two genes for each monomer”.

Response:

We have changed the sentence which now reads “The bicistronic plasmid carries a single promoter followed by a gene for each monomer ...” at line 144-145.

6) It is surprising and notable, and critical to success of the present design, that nearly exclusive formation of a heterodimer is obtained from bi-cistronic expression. What prevents homodimer formation? There doesn't appear to be any energetic/thermodynamic contribution, so does this have something to do with the tendency of the two chains coming sequentially off the same ribosome to co-assemble? What do the authors suggest is going on here. It is very unusual.

Response:

We are able to work with pure heterodimer due to careful purification. In concept, we expect three species of dimers, Cp149_{His} homodimer, Cp149_{Y132A} homodimer, and Cp149_{His}Cp149_{Y132A} heterodimer. Cp149_{His} homodimer can assemble into capsids in *E. coli*, but the other two stay as dimers. During purification we excluded capsids via size

exclusion chromatography, removing Cp149_{His} homodimer while retaining Cp149_{Y132A} homodimer and Cp149_{His}Cp149_{Y132A} heterodimer. Then we use Ni chromatography to isolate Cp149_{His}Cp149_{Y132A} heterodimer which has a His-tag while eluting Cp149_{Y132A} homodimer in the flow-through.

We have added on a sentence to state how the two types of homodimers were removed during purification at line 147-149.

7) Should CDMS be described as a variation on a native mass spec method to help the reader understand that the masses of intact, non-covalent assemblies are being measured?

Response:

In the CDMS instrument developed in the Jarrold lab, an aqueous sample is introduced by electrospray ionization. In short, it is a native sample. We have modified the text appropriately in methods (line 96) and results (line 177).

8) The paper talks in several places about the capsid getting “refilled” after removal of the special patch. This needs to be rethought, since nothing is actually going inside the capsid. Maybe the shell is being re-annealed, or re-assembled, or re-sealed, or repaired? Or maybe the *surface* of the shell is being refilled.

Response:

We have systemically changed refilled to variants of “refilled the surface” with a few exceptions.

9) The authors need to be a bit more circumspect in how they talk about the generalizability of what has been done here. The present approach relies on a system with *multiple levels* of hierarchical assembly control; the dimer must be preparable separate from the hexamer of dimers, and likewise for the hexamer of dimers. And a rather narrow thermodynamic window has to be hit by mutagenesis in order to get the whole system to work. It’s a testament to the difficulty of what was accomplished here, and the unlikelihood (in this reviewer’s opinion) of extending it to very different systems doesn’t take away from the current success. The authors should be more specific about what would be required to extend the ideas to other systems.

Response:

As outlined in the following paragraph, we feel our approach is generalizable. There are several systems where manipulating assembly and disassembly to make holey capsids is probably within reach – Cowpea Chlorotic Mottle Virus, bacteriophage MS2, and bacteriophage P22 come to mind – but we agree with the reviewer that we should describe some of the requirements for the system needed to make this experiment work.

In the new penultimate paragraph of the conclusion we now write:
Many viral capsids have subunits that can be asymmetricized, though it is easiest to see doing this with monomers and dimers. We anticipate that our strategy to create an asymmetric assembly pathway and asymmetric capsids can be generalized to some other capsid-like systems. There are likely to be some fundamental requirements needed to thermodynamically separate the individual steps of the reactions. First, the self-assembly system must be reversible. Second, it must be possible to create an asymmetric subunit where at least one face is able to conditionally stop assembly under conditions where the other face assembles. In HBV, the Y132A mutation provides an additional barrier to assembly that might have proceeded once a nucleus is formed, even at low ionic strength. Third, it must be possible to overcome the conditional stop. In HBV we depended on the multivalent binding sites afforded by the hexameric nucleus and the relatively high association energy for subunit addition afforded by high ionic strength. Finally, there must be a substantial energy gradient between the nucleus and the growing capsid that can be manipulated to change which domain is more stable. In low ionic strength and moderate Ni^{2+} concentration, the hexameric nucleus is more stable than capsid. After disulfide crosslinking and in the presence of EDTA and urea, the Cp150 capsid is far more stable than the hexamer. Engineering a crosslinkable His-tag and a crosslinkable cysteine into a location near a vertex would seem a good starting strategy. We note the use of elastin-like peptides in Cowpea Chlorotic mottle virus capsid protein as an example of a capsid with two modes of regulatable assembly⁴.

10) On a similar point, the authors indicate that “not every viral capsid is built from homodimers”. That’s an understatement! It’s rather rare, so this needs to be rephrased, and if there are other candidate capsids based on similar hierarchical assembly of dimers then that more specific information would be useful to readers.

Response:

The reviewer is correct. One of us (AZ) has spent too much time thinking about small, icosahedral, RNA plant viruses (e.g. bromoviruses, tombusviruses, sobemoviruses) where dimers are relatively common. The sentence has been modified and is now the opening sentence of the paragraph above.

11) In describing prior work on engineering protein capsids and cages, the authors should consider rephrasing in line 43 where they cite what other authors have “attempted”. Given the successes that have been achieved by the cited papers in this area, a different word choice might make sense: “...have exploited the advantages of natural systems...”?

We appreciate the modification that reviewer proposed. We have changed “attempted to” to “exploited” in our text (line 47).

Author's Response to Reviewer #3:

Zlotnick and colleagues describe a hierarchical assembly of asymmetric HBV capsids and use an impressive array of analytical techniques (cryoEM, CD-MS and RPS) to characterize the products.

The cp149-Y132A constructs and its co-assembly with 'WT' cp149 has been described before in refs 34-36. Here, novel constructs for formation of a cp149//cp149-Y132A heterodimer is described. In addition, a procedure to assemble a patchy particle, using hexamers and double hexamer seeds composed of the heterodimer, is described. It is shown that this heterodimer patch can be selectively removed and subsequently filled out with additional building blocks.

1) In the abstract of the manuscript, the authors claim that 'this strategy can be generalized', but no explanation or demonstration of this bold claim is given. It seems this strategy is actually rather specific to HBV, and only applicable to capsids with dimeric building blocks, with a nucleation-elongation type assembly pathway.

Response:

This point is now discussed in detail in the new penultimate paragraph of the conclusions. Please see the detailed response to Reviewer 2's question 9.

2) In the introduction the authors state that a current shortcoming in the field of virus-based nanoplatfoms is that asymmetric particles cannot yet be assembled, limiting applications. A discussion is missing about which particular applications would require such features.

Response:

A specific response to this question now makes up the last two paragraphs of conclusions, particularly the last paragraph. Please also see also reviewer 1 question 2 and reviewer 2 question 9.

3) It is not yet entirely clear to me if the procedure yields 'pure' composite particles, or also contains a substantial fraction of full (wt cp149) homodimer particles. On that note, the complete absence of any homodimers in figure 2c is remarkable and deserved some additional discussion/explanation.

Response:

We think the reviewer asked a fair question whether we have "pure" composite particles during co-assembly. Our goal was to create hexamer nucleated capsids and minimize the fraction Cp150 self-assembled capsids. When we made holey capsids by disassembly, we observed leftover T=4 and T=3 capsids by RPS and cryo-EM. Although we do not have a definite answer whether those leftover T=3 and T=4 capsids

are “pure” capsids, we suspect that majority of these capsids are Cp150 self-assembled capsids. And that is the reason that we cannot disassemble them into holey capsids.

For the purification of heterodimer, please see response to reviewer 2 question 6.

4) Another point that is not clear is how the T=3 particle is formed in the hierarchical assembly procedure as the manuscript almost entirely focusses on the T=4 structure. The RPS data in figure 4b cuts out the region below T=3 that might show any defected particles. Could these same distributions also be assessed by CD-MS for defects in both T=3 and T4 particles? In any case, the RPS data suggests that removal of heterodimers from assembled particles results in much more heterogeneous defects than only hexamers/double hexamers.

Response:

A discussion of T=3 particle spontaneous assembly is now included in the manuscript and in detailed responses to reviewers 1 and 2. In the full spectrum from RPS experiments, no fragments smaller than 90-mers corresponding to T=3 capsids were observed. In dissociation and refilling experiments, the relative amount of T=4 particles decreased and reverted, respectively; the amount of T=3 particles, normalized for total counts, remained constant. This suggests that T=3 particles assemble without hexamers. Consistent with this hypothesis, we have observed that T=3 assemble early in the reaction^{2,5}, before T=4 particles accumulate, and then no further T=3 particles form. This suggests that T=3 particles have a peculiar nucleation event and arise from spontaneous assembly without need for a hexameric nucleus.

We opted to use RPS over CDMS for these samples. RPS works best at high ionic strength and is otherwise agnostic to most solutes. CDMS of these samples would have required substantial preparation as they included high concentrations of urea, a CDMS-unfriendly solute. CDMS requires volatile buffers such as ammonium acetate. Thus, RPS was able to rapidly evaluate samples that required little or no additional purification. These are two complementary techniques.

5) On p9 line 153, it is written that the constructs could tolerate larger insertions for antigen presentation, but no such data is shown.

In the literature⁶ and by personal communication, several efforts to add large inserts to the spike tips have failed, probably because of collisions between the inserts at the intra-dimer interface. We speculate that an asymmetric dimer offers the opportunity to avoid intra-dimer collisions. At this time, we do not have evidence to test our hypothesis. That will be a future series of experiments. The goal of this paper is to show readers that HBV heterodimer and holey capsids are achievable and to present our new assembly pathway and platform. Many works have been successfully done by other groups, specially the Nassal group, showing that large proteins, like GFP and OspA, can be conjugated genetically to the HBV capsid surface. We believe that our new

heterodimer provided a new choice, specifically presenting one copy of insertion on a heterodimer instead of two copies on a single homodimer. Overall, we anticipate that all the successful protein insertions that have been accomplished by other groups can be transferred to our new heterodimer.

6) On p10 line 172, the resolution estimate of the negative stain model is in my opinion a bit inflated, and reported with too much precision (is it 17 or 18 angstrom? or actually just about 2 nm?).

Response:

The data are negative stain, which provide a surface contour of this relatively small complex. The observed FSC was 17Å at a correlation of 0.143 and 25Å at a correlation of 0.5. However, we note that the reported resolution based on the FSC calculation may be influenced by the self-consistent molecular envelope of the stain, distorting the resolution determined using the gold-standard method in Relion. Therefore, we agree with the reviewer that it may overestimate the “true” resolution. Calculated density, low-pass filtered to the listed resolutions shows little differentiation at this resolution range (see below). We have changed the text as follows (line 185): The top 8 classes were selected for reconstructing a 3D density map that reached a resolution of approximately 2nm.

7) On p10 line 178, it is written that different sizes of assembly seeds can be formed with other mutations, but no evidence of this is given, nor any explanation.

Response:

This speculation was removed as the reviewer's suggested. It is a future line of study for us.

8) On p13/14, the discussion of the size of the hole in the cryoEM reconstructions is

over interpreting the densities in my opinion. I don't think that hexamer or double hexamer defects are effectively sorted in the image classification, or that individual images are 100% accurately aligned, and that therefore the apparent size of the hole in the map is a convoluted effect of these image processing artefacts, sample heterogeneity and the actual physical size of the hole in the capsids.

Response:

We agree with the reviewer that the sample is heterogeneous, which was observed in the RPS experiment. We also agree with the reviewer that even under the ideal imaging conditions, not all the images can be 100% accurately aligned. We describe results that are consistent with observed data. We sorted images into two pools based on the size of an aligned hole and the resulting reconstructions represent averages. We have acknowledged these facts in the text: "We note that some capsids included in the complete capsid dataset may have had holes that were obscured by particle orientation" and "The two holey capsid density maps never reached the same resolution as the 120-dimer capsid density map. We believe that might be due to the use of fewer particles, particle heterogeneity, and potentially the greater flexibility of holey capsids."

However, our data show that in holey capsids, (i) holes have consistent features and (ii) their locations with respect to the icosahedral symmetry of an intact capsid are also consistent. If the defects in the images fit a normal distribution, in a reconstruction they would contribute noise over the entire 3D map. If the holes could not be aligned because they were random in shape or were randomly distributed, we would expect to see a smooth but complete capsid as shown in observations and model calculations that we reported previously (*Protein Science* **26** (11), 2170-2180, *Viruses* **10** (1), 25).

Furthermore, noise in the hole is similar to noise outside the capsid envelope. At the contour level where the remaining HBV spikes contribute to the full mass, we estimated ~9 dimers and ~18 dimers missed from the 3D maps, respectively. If we keep increasing the contour level, we observed degraded density close to the edge of the hole; it shows dimers at those locations are heterogeneous or prone to be exposed to the reaction. The observed sizes for the holes are consistent with RPS.

Taken together, though the number of missing dimers from the holey capsid is heterogeneous, our data are consistent with a population of capsids with a small hole based on a single missing hexamer and a larger hole consistent with two missing hexamers. In both cases, a distribution of peripheral subunits is also missing. We do not know if the larger hole is derived from a double hexamer nucleus or a single hexamer.

9) In figure 3b, the mass of the assigned double hexamer is wrongly annotated and I also believe that the assignment of that signal as being a double hexamer is somewhat wishful. What would the theoretical mass of the annotated complexes be and how accurate are the experimental masses?

Response:

We thank the reviewer for catching our mis-labelling. Now we have corrected Fig 3b. The calculated mass for a heterodimer is 34271 Da. A hexamer should be 205,626 Da and a double hexamer, which has 11 dimers, should be 376,981 Da.

In CDMS, there are essentially no false positive signals. Each ion is trapped, oscillating, for 100ms while measuring the frequency (proportional to m/z) and amplitude (proportional to z) of the signal. Small errors can arise in the measurement of m/z and z resulting in the peak width in the spectrum. An experimentally more important source of variability is that water and salts can associate with large ions.

The hexamer mass we determined using CDMS agrees well with the calculated mass. The putative double hexamer signal is a smear beginning at about 375 kDa and peaking at 391 kDa, suggesting an affinity for water and salts. However, in support of this assignment, and we observe the presence of a small population of double hexamers by EM (Supplementary Fig. 3). Additional information has been added at line 177-181 to clarify the expectations of the technique.

10) In Figure 4c, the lower fluorescence signal of the cp150Bo capsids is interpreted as a fluorescence quenching effect. Can the authors exclude that no quenching takes place, but that the fluorescence signal of the assembled capsid is simply lost against the high background of free dimers?

In response to reviewer 1's question 3, we have added new data (supplementary figure 10) showing the relative amounts of Cp150Bo in different samples by absorbance. These data show that the surface-refilled holey capsids have about the same amount of Bo absorbance at 504nm as the pure Cp150Bo capsids. However, the fluorescence of the surface-refilled holey capsids is substantially greater than observed for capsids of pure Cp150Bo (Fig. 4c). These data were recorded with an HPLC outfitted with a diode array absorbance detector and a fluorescence detector to facilitate these sorts of measurements.

- 1 Stray, S. J., Johnson, J. M., Kopek, B. G. & Zlotnick, A. An in vitro fluorescence screen to identify antivirals that disrupt hepatitis B virus capsid assembly. *Nat Biotechnol* **24**, 358-362 (2006).
- 2 Asor, R., Schlicksup, C. J., Zhao, Z., Zlotnick, A. & Raviv, U. Rapidly Forming Early Intermediate Structures Dictate the Pathway of Capsid Assembly. *J Am Chem Soc* **142**, 7868-7882, doi:10.1021/jacs.0c01092 (2020).
- 3 Lee, L. S. *et al.* A molecular breadboard: Removal and replacement of subunits in a hepatitis B virus capsid. *Protein Sci* **26**, 2170-2180, doi:10.1002/pro.3265 (2017).
- 4 van Eldijk, M. B. *et al.* Designing two self-assembly mechanisms into one viral capsid protein. *J Am Chem Soc* **134**, 18506-18509, doi:10.1021/ja308132z (2012).
- 5 Harms, Z. D., Selzer, L., Zlotnick, A. & Jacobson, S. C. Monitoring Assembly of Virus Capsids with Nanofluidic Devices. *ACS Nano* **9**, 9087-9096, doi:10.1021/acsnano.5b03231 (2015).
- 6 Bottcher, B., Vogel, M., Ploss, M. & Nassal, M. High plasticity of the hepatitis B virus capsid revealed by conformational stress. *J Mol Biol* **356**, 812-822 (2006).

REVIEWERS' COMMENTS

Reviewer #1 (Remarks to the Author):

I would like to thank the authors the revised version of the article. They have address most of the questions raised in a satisfactory manner, in particular regarding SEC experiments. However, the authors should address some minor issues:

Question 3: The CP150Bo content determined by SEC is in good agreement with cryo-EM and RPS. However, in order to yield a meaningful value of population ratios, the peak area should be employed instead the absorption value at 7.5 mL.

Question 7: I agree with the authors that the capsids are morphologically normal, as shown in Supplementary Figure S4 and S5.

However, from these experiments, no information about CP segregation can be obtained. The gel provided as reply to this question is the first experiment designed to prove this (line 216-217). Why is this figure not included in the article? It will give some experimental grounds for such statement.

Additionally, it would be helpful that the lines referred in the responses will correspond to the actual statements.

Reviewer #2 (Remarks to the Author):

The revised version of the manuscript is clarified and improved -- now ready for publication.

Reviewer #3 (Remarks to the Author):

In the revised manuscript, Zlotnick and colleagues have addressed and clarified all technical issues I highlighted in the original review regarding the role of the T=3 particle, mass assignments in CD-MS spectra, EM resolution estimates, and fluorescence quenching effect of the cp150-Bo particles.

My main concern with the original manuscript was in the bold claims made regarding the generalizability and potential applications of the hierarchical capsid assembly for antigen presentation. The authors have argued that this will be the subject of future work and have done little to either demonstrate those applications in the current work, or dial down the claims that are made. The applications and generalizations that the authors highlight are easier said than done and thereby remain a big promise that is currently left unfulfilled.

Author's Response to Reviewer #1:

I would like to thank the authors the revised version of the article. They have address most of the questions raised in a satisfactory manner, in particular regarding SEC experiments.

Response:

We thank the reviewer for approving our responses.

However, the authors should address some minor issues:

Question 3: The CP150Bo content determined by SEC is in good agreement with cryo-EM and RPS. However, in order to yield a meaningful value of population ratios, the peak area should be employed instead the absorption value at 7.5 mL.

Response:

We chose to look at a narrow slice of the chromatograph to ensure we were comparing apples to apples, that is complete T=4 capsids to complete T=4 capsids. Particularly in the trailing edge of the chromatographic peak, we observe that the samples are enriched for T=3 particles, which would result in an underestimate of the extent of refilling.

Question 7: I agree with the authors that the capsids are morphologically normal, as shown in Supplementary Figure S4 and S5.

However, from these experiments, no information about CP segregation can be obtained. The gel provided as reply to this question is the first experiment designed to prove this (line 216-217). Why is this figure not included in the article? It will give some experimental grounds for such statement.

Response:

We appreciate this point made by the reviewer. With respect to the overall story, we believe that our later cryo-EM data are more definite for us to draw the conclusion that we have segregation. And also, we think we have already overloaded our main manuscript with figures. Therefore, we prefer to leave the gel image in supplementary information will be enough.

Additionally, it would be helpful that the lines referred in the responses will correspond to the actual statements.

Response:

We apologize for our mis-labelling.

Author's Response to Reviewer #2:

The revised version of the manuscript is clarified and improved -- now ready for publication.

Response:

We truly appreciate this comment from the reviewer.

Author's Response to Reviewer #3:

In the revised manuscript, Zlotnick and colleagues have addressed and clarified all technical issues I highlighted in the original review regarding the role of the T=3 particle, mass assignments in CD-MS spectra, EM resolution estimates, and fluorescence quenching effect of the cp150-Bo particles.

Response:

We appreciate the efforts of the reviewer.

My main concern with the original manuscript was in the bold claims made regarding the generalizability and potential applications of the hierarchical capsid assembly for antigen presentation. The authors have argued that this will be the subject of future work and have done little to either demonstrate those applications in the current work, or dial down the claims that are made. The applications and generalizations that the authors highlight are easier said than done and thereby remain a big promise that is currently left unfulfilled.

Response:

We understand the concern from the reviewer. Now we have truncated our abstract, removing speculation regarding application. We also deleted our suggestion of antigen presentation to avoid controversies. We will test our system for such applications in our future work. Though we are more cautious than in our initial manuscript and are aware of the hurdles in development, we feel strongly that we have provided a generalizable outline for developing hierarchical assembly in other virus and virus-like assembly systems.